# Research on the Corrosion Resistance of an Epoxy Resin-Based Self-Healing Propylene Glycol-Loaded Ethyl Cellulose Microcapsule Coating

Shudi Zhang [1,*], Linkun Liu [1], Yuheng Xu [1], Quanda Lei [1], Jiahui Bing [1] and Tao Zhang [2,*]

[1] School of Environmental and Chemical Engineering, Shenyang Ligong University, Shenyang 110159, China; 15670291571@163.com (L.L.); lovexyh2023@163.com (Y.X.); lqd317213741@163.com (Q.L.); 18342029676@163.com (J.B.)
[2] Chinese Academy of Sciences (Shenyang) Metals Research, Shenyang 110016, China
* Correspondence: zhangshudi@163.com (S.Z.); zhangtao@mail.neu.edu.cn (T.Z.)

**Abstract:** In this work, ethyl cellulose was used as a wall material, propanetriol as a core material, polyvinyl alcohol as a stabilizer and gelatin as an emulsifier. Self-healing microcapsules with a slow-release effect were prepared using the solvent evaporation method. Various analytical techniques, such as 3D confocal microscopy (LCSM), optical microscopy (OM), scanning electron microscopy (SEM), infrared spectroscopy (FT-IR), energy dispersive spectroscopy (EDS), thermal weight loss analysis (TGA), laser particle size tester and electrochemical impedance polarization, are utilized. The morphology, distribution, particle size, corrosion resistance and self-healing ability of the prepared microcapsules and resin-based coatings were characterized and analyzed. The results show that the cross-sectional core–shell structure is clearly seen in the LCSM, showing a smooth, hollow, spherical shape. OM and laser particle size testers have shown that the size of the microcapsules decreases over time. Also, in OM, the microcapsules are uniformly distributed in the emulsion with a smooth and non-adherent surface. In SEM, the microcapsule particle size is about 150 μm, the shell wall thickness is about 18 μm, and the hollow structure of ruptured microcapsules is obvious. FT-IR and TGA confirmed the successful encapsulation of the formulated microcapsules. The results show that when the core-wall mass ratio is 1.2:1 and the amount of microcapsule is 10% of the coating amount, the prepared microcapsule has high thermal stability and certain wear resistance. By electrochemical and immersion experiments, it was found that a 3.5 wt % NaCl solution has the best impedance, the lowest corrosion current density, and good adhesion and tensile toughness. The results showed that glycerol was successfully released from the broken microcapsules and self-healed, forming an anticorrosive coating with excellent corrosion resistance and self-healing ability.

**Keywords:** microencapsulation; E-51 epoxy resin; self-healing; corrosion resistance; AZ91D magnesium alloy

## 1. Introduction

Magnesium alloy has shown high application value in various applications in today's society. In the field of lightweight alloys, it is of great significance as a benefit mankind and to the promotion of further developments in science and technology. Magnesium alloys are also the lightest structural metal materials in the world [1–3]. They have the advantages of having a high specific strength sufficient toughness, and being lightweight and easy to transform, hence their pivotal in the development of society from ancient times to the present [4–7]. Therefore, magnesium alloys play an indispensable role in the fields of construction, aerospace, smart wearable, portable home, and so on, as well as in the energy and chemical industries. [5–8]. However, China is a large consumer of magnesium resources, it is not difficult to predict that under the development trend of today's world, further research and application of magnesium alloys will increase in significance [6–10].

Therefore, in this context, the poor corrosion resistance of magnesium alloy itself restricts the development of magnesium alloys. This situation urgently needs to be addressed [11–14]. The surface of magnesium alloys is similar to that of aluminum alloys; it also has a layer of magnesium oxide coating. However, the coating itself is loose, porous, soft and thin, and it is difficult to provide good corrosion protection [15–19]. This requires a layer of organic coating on the surface to prevent corrosion and improve its utilization. However, considering that it is not easy to dispose of the coating after the application or recycling of magnesium alloys, it is necessary to consider more environmentally friendly, highly efficient and long-lasting corrosion-resistant coatings [20–24]. In this context, coatings with self-repairing abilities and added microcapsules have emerged.

In recent years, self-healing microcapsules have achieved remarkable advantages in terms of biocompatibility and environmental friendliness, which has attracted much attention in various fields, such as materials chemistry, biopharmaceuticals and environmental sciences [25–29]. In particular, microencapsulation technology, with a core–shell structure and the synergistic effect of self-healing and slow release, has started to be explored for the corrosion and protection of alloy surfaces [30–34]. This is because microcapsules act as fillers in the preparation process to provide a good slow-release refinement and significantly improve the coating microstructure [34,35]. The self-healing ability of microcapsules is a very promising method for repairing microgaps in epoxy-resin-based magnesium alloy materials. There are two general types of self-healing microcapsules: One is the general type of microcapsule, and the other is the slow-release type of microcapsule. White SR [36] et al. attempted to design a general-type self-healing microcapsule by means of in situ polymerization. The microcapsules were composed of polyurea-formaldehyde resin-coated dicyclopentane, which is a general microcapsule of the resin-coated dicyclopentadiene type. The slow-release self-healing microcapsules are released from the wall material by controlling the core material within the core–shell structure. They can be used in alloy corrosion protection, chemical production as well as in the agricultural and food industries. Chen [37] designed a structurally robust microcapsule with a slow-release synergistic effect using a face complex of mesoglycan and chitosan from pectin as well as latex proteins. Cunha [38] studied bifunctional polyurea-formaldehyde microcapsules and successfully prepared microcapsules by mixing linseed oil and benzotriazole. Wang [39] prepared slow-release microcapsules that can act as a vitamin supplement using spray technology, while the natural antioxidant capacity and release rate of the microcapsules were tested for modulation. Amand [40] prepared a capsule structure with a synergistic effect of self-healing and slow release by varying the acetone content. The generated content of the core wall was also investigated, and when the microcapsules produced an effect, the core material flowed out, which in turn diffused to fill and bind the cracks.

Compared to existing literature reports, the uniqueness of this research lies in the fact that PVA itself has bonding, emulsification and dispersion properties, while gelatin itself is a macromolecular hydrophilic colloid, which can also be used for bonding. Therefore, the simultaneous addition of poly (vinyl alcohol) and gelatin provides stabilization as well as enhances aqueous phase viscosity to form a thin film, thereby increasing the encapsulation rate of the microcapsules. The wall material, ethyl cellulose, has the functions of bonding, filling and coating forming. However, there is limited research on the emulsifiers used in compounding. We analyzed the morphology of the microcapsules using a single emulsifier, as well as the lack of a certain heart–wall ratio and optimal mass fraction between the entire composition and the coating. This is the focus of this work. Finally, LCSM was used to observe the macroscopic morphology of the microcapsules, and the existence of core–shell structure was preliminarily determined. The particle size distribution and dispersion uniformity of the microcapsules in emulsion were evaluated using OM and a laser particle size analyzer. The micromorphologies of the microcapsules and the self-healing coatings were observed by scanning electron microscopy. In addition, FT-IR and TGA were used to verify that the microcapsules were successfully encapsulated. The basic properties of microcapsule coatings are tested via electrochemical Nyquist diagrams and Tafel

analyses, immersion experiments, adhesion tests and coating tensile tests. The purpose of this experiment was to show that the self-repairing coating with added microcapsules has excellent self-healing properties, corrosion resistance characteristics, adhesion, and self-healing tensile ductility under the action of a compounded emulsifier. Meanwhile, the impedance value in 3.5 wt.% NaCl solution was $8.242 \times 10^4$, which reached the optimum value. And it had the lowest corrosion current density and good adhesion and tensile toughness. Through the above corresponding series of experiments, it was able to be thoroughly analyzed that the glycerol was successfully released from the broken microcapsules for self-heal. The AZ91D magnesium alloy anticorrosive coating with excellent corrosion resistance and self-healing ability was formed. Compared to other experiments, the wall material, ethyl cellulose, itself is easy to degrade, green and has good biocompatibility as well as the function of bonding and filling. Finally, new theoretical support and empirical methods are provided for the corrosion and protection of magnesium alloy surfaces in this field.

## 2. Materials and Methods

### 2.1. Materials and Instruments

Main raw materials: AZ91D magnesium alloy (density 1.82 g/cm$^3$); ethyl cellulose (density 1.07 g/cm$^3$); propanetriol (relative molecular mass 92.09, colorless, odorless, sweet, clear and viscous liquid appearance; Sinopharm Chemical Reagent Co., Shanghai, China); polyvinyl alcohol (relative molecular mass 44.05, white flake, flocculent or powdery solid); Shanghai Sinopharm Chemical Reagent Co., Ltd., Shanghai, China); gelatin (also known as animal gelatin, gelatin belongs to a large molecule hydrocolloid; Tianjin Damao Chemical Reagent Factory, Tianjin, China); OP-10 (white and milky white paste; Shanghai Aladdin Biochemical Technology Co., Ltd., Shanghai, China); dichloromethane (relative molecular weight 84.93, colorless and transparent liquid; Tianjin Damao Chemical Reagent Factory, Tianjin, China); n-butanol (relative molecular weight 74.12, colorless and transparent liquid; Tianjin Damao Chemical Reagent Factory, Tianjin, China); xylene (relative molecular weight: 106.165; Tianjin Damao Chemical Reagent Factory, Tianjin, China); anhydrous ethanol (colorless liquid, with special fragrance; Tianjin Damao Chemical Reagent Factory, Tianjin, China); and E-51 Epoxy resin and supporting curing agent (Zhongtian Fine Chemical Co., Ltd., Zhoushan, Zhejiang Province).

Main experimental instruments: HH-S2 digital thermostatic water bath (Changzhou Yineng Experimental Instrument Factory, ChangZhou, China); DZF vacuum drying oven (Nanjing Suenrui Production Plant, Nanjing, China); 85-2 digital thermostatic magnetic stirrer (Changzhou Jintan Jingda Instrument Manufacturing Co., Ltd., Changzhou, China); CHINALAB 20 electronic balance (Anhui Tianping Machinery Co., Ltd., Chizhou, China); CHI660E-type electrochemical workstation (Gongyi Kerui Instrument Co., Ltd., Gongyi, China); VEGA3 XMU scanning electron microscope (Guangzhou Dongrui Technology Co., Ltd., Guangzhou, China); 3D confocal microscope (Xiamen Maina Optical Technology Co., Ltd., Xiamen, China); NMM-800TRF optical microscope (Dongguan Ruixian Optical Instrument Co., Ltd., Dongguan, China); FBS-50KNW tensile tester (Jinan Tianhua Testing Equipment Co., Ltd., Jinan, China); GD26-FTIR-650 Fourier transform infrared spectrometer (Zhongke Ruijie (Tianjin) Technology Co., Ltd., Tianjin, China); and TGA55 thermal weight loss analyzer (Nanjing Nanda Wanhe Technology Co., Ltd., Nanjing, China). The main experimental instruments are shown in Figure 1 below.

### 2.2. Sample Preparation

An AZ91D magnesium alloy with the size of 15 mm × 15 mm × 25 mm was selected And sanded with 600#, 800#, 1500# and 2000# sandpaper, in this order, until a metallic luster appeared. After, it was rinsed with deionized water for 3 minutes and put into a beaker filled with absolute ethanol. Ultrasonic shaking treatment was carried out for 5 min for the purpose of de-oiling. It was then removed, rinsed again with deionized water for 2 min, and put in alkaline washing solution. After rinsing with deionized water for 2 min,

it was degreased with acetone. Then, it was removed and rinsed with deionized water for 3 min and put in an acid washing solution. After rinsing with deionized water for 3 min, it was put into the vacuum drying oven and dried at 50 °C for use.

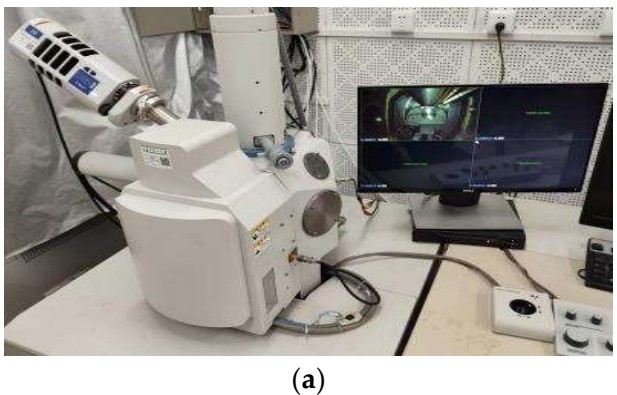
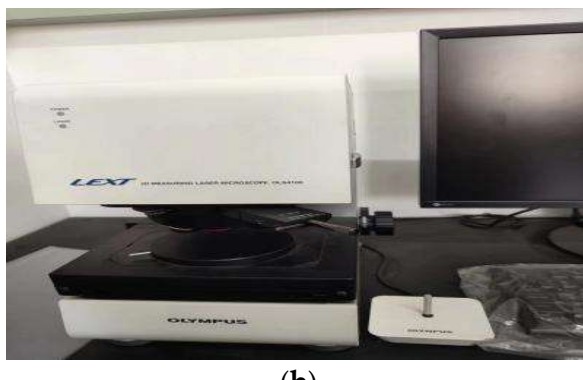

(**a**)  (**b**)

**Figure 1.** Macroscopic view of the main experimental instruments: (**a**) scanning electron microscope; (**b**) 3D confocal microscopy (LSCM).

### 2.3. Experimental Content

2.3.1. Preparation of Microcapsules

Stirring was continued at 200 rpm for 5 min to obtain 250 mL of 0.8 wt.% PVA with 1 wt.% gelatin as a microencapsulated stabilizer. Next, 20 g of ethyl cellulose was fully dissolved in 100 mL of dichloromethane. 8 mL of propanetriol was added to the mixture of ethyl cellulose and dichloromethane. It was mechanically emulsified at 600 rpm in a constant temperature water bath at 45 °C and fully reacted for 4 h until the dichloromethane was completely evaporated. After the reaction, the microcapsules were isolated by filtering and rinsing with deionized water and were dried in a vacuum drying oven at 60 °C for use.

2.3.2. Preparation of Self-Healing Coatings

In the process of adding microcapsules into the coating, the influence of the amount and dispersion of microcapsules on the properties of the coating was researched. The designs are no add, 5% addition, 10% addition, and 20% addition. Twelve 25 mL beakers were taken and divided into three groups, a, b and c, resulting in three groups of 3 beakers each. 3.75 g of epoxy E-51 was put in each beaker. 1.25 g of hardener b was put in a beaker. The organic solvent in beaker c was put into a and b in an appropriate amount and stirred with a magnetic stirrer to mix thoroughly. Among them, 0 g, 0.5 g, 1 g, 2 g were added to each component and stirred for 30 min using a magnetic stirrer. Finally, the epoxy resin self-healing coating containing microcapsules with different mass fractions was uniformly coated on the surface of the magnesium alloy and then cured at 25 °C for 12 h and then at 60 °C for 24 h in a vacuum drying box.

### 2.4. Testing and Analysis

2.4.1. Electrochemical Test

The electrochemical impedance spectrum and polarization curves of the self-healing coatings were obtained using a CHI660E electrochemical workstation. The experimental tests were performed using a conventional three-electrode system. The specimen was the working electrode (exposed area of 1 cm$^2$), the saturated glycerol electrode (SCE) was the reference electrode, and the platinum electrode was the auxiliary electrode. The test solution was 3.5% N saturated aCl solution, and the test temperature was (25 ± 5) °C. The electrochemical impedance spectrum was scanned from 0.01 Hz to 100,000 Hz, the polarization curve was scanned at a voltage range of ±0.5 V relative to the open circuit potential, and the scan rate was 5 mV/S. To ensure accuracy and reproducibility, three sets of parallel tests were performed for each test.

### 2.4.2. Scanning Electron Microscopy and X-ray Diffraction Spectroscopy

A VEGA3 XMU scanning electron microscope (SEM) (Guangzhou Dongrui Technology Co., Ltd., Guangzhou, China) from TESCAN, with an accompanying energy spectrometer (EDS), was used to observe the prepared microcapsules as well as the microscopic morphology and microscopic characterization of the coatings, with a scanning voltage of 20 kV.

### 2.4.3. Microcapsules Particle Size and Macroscopic Morphology Analysis

3D confocal microscope (LCSM) (Xiamen Maina Optical Technology Co., Ltd., Xiamen, China), optical microscope (OM) (Dongguan Ruixian Optical Instrument Co., Ltd., Dongguan, China), and laser particle size tester (Kunshan Lugong Precision Instrument Co., Ltd., Kunshan, China) were used to characterize the macroscopic morphology, distribution in the compounded emulsion, particle size, and the aggregation of the prepared microcapsules. Among them, LCSM uses a 405 nm laser as the light source to initially determine the microcapsule roughness and core–shell structure.

### 2.4.4. Fourier Infrared Spectroscopy and Thermal Weight Loss Testing

The GD26-FTIR-650 Fourier infrared spectrometer (Zhongke Ruijie (Tianjin) Technology Co., Ltd., Tianjin, China) was used to view the organic functional group profile of the microcapsules. Among them, about 0.3 mg KBr was weighed and fully ground with microcapsules in an agate mortar, pressurized to 15–20 MPa and kept for 1 min for testing. The TGA55 heat loss tester (Nanjing Nanda Wanhe Technology Co., Ltd., Nanjing, China) was used for testing, and the sample was tested by opening the pure nitrogen valve and adjusting the outlet pressure to 0.1 MPa.

## 3. Results and Discussion

### 3.1. Characterization of Microcapsules

### 3.1.1. Microstructure and Particle Size Distribution

Figure 2 shows the SEM morphology and particle size distribution of the microcapsules and microstructure. Figure 2a shows the SEM image of microcapsules loaded with glycerol at 500 μm. It can be seen from the figure that the microcapsules are relatively uniform in both distribution and size. Local rupture appears, and the core–shell structure can be seen initially from the rupture. Figure 2b shows the SEM image of the microcapsules at 200 μm, and further magnification shows that the surface of the microcapsule is smooth and round and that the size distribution is relatively uniform. Figure 2c shows the SEM image of a single microcapsule, which is the moment when the magnification could be maximized and a clear image could be taken under the right conditions. At 100 μm, the synthesized capsule appears spherical, then becomes very round and smooth again. The core and wall states of the core–shell structure can be seen under reflection, in addition to the compact distribution of the capsule-forming structure. Figure 2d shows the SEM cross-section of the microcapsules rupture. The total thickness of the outer and inner walls of the capsule is about 15 μm, and Figure 2e shows the SEM image at a magnification of 100 μm. At this time, not only can the core–shell structure be clearly seen, but also the change distribution of microcapsule particle size can be seen. The observed results correspond to the particle size distribution detected by the laser particle size tester. Figure 2f shows the particle size distribution of the microcapsules, ranging from (150.424 ± 3.756) μm, which is consistent with the Gaussian nonlinear fitting results.

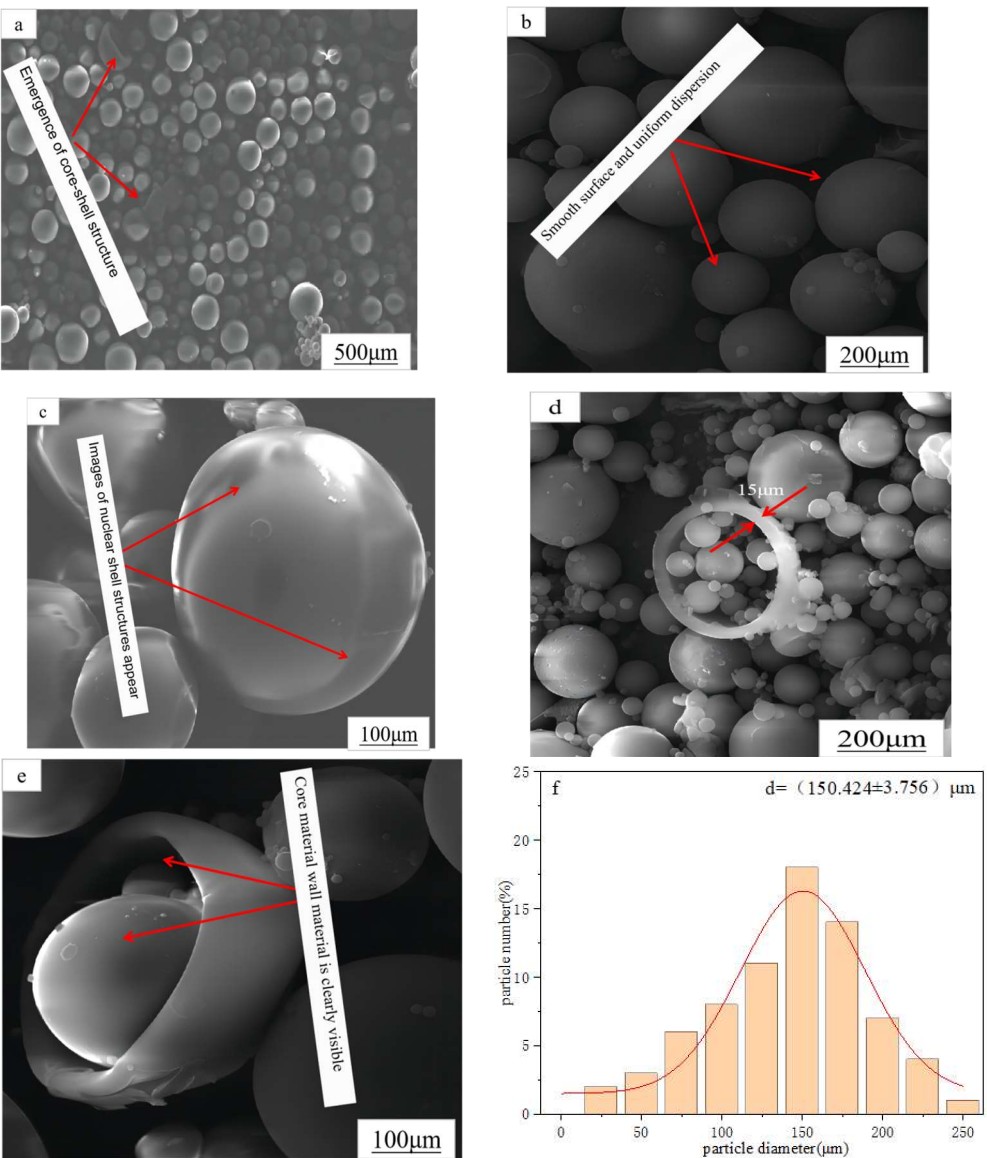

**Figure 2.** SEM images and size distribution of glycerol-loaded microcapsules: (**a**) microcapsule magnification at 500 μm; (**b**) microcapsule magnification at 200 μm; (**c**) microcapsule magnification at 100 μm; (**d**) ruptured microcapsule magnification at 200 μm; (**e**) ruptured microcapsule magnification at 100 μm; (**f**) size distribution of particle size.

### 3.1.2. Macroscopic Structure and Particle Dispersion

Figure 3 shows the macroscopic structure and particle size morphology of the microcapsules prepared under the 3D confocal microscopy (LCSM), optical microscopy (OM), and camera 2a. Figure 3a,b show the morphology of the capsule emulsion prepared under optimal conditions and the capsule emulsion dried in a vacuum drying oven, respectively. It can clearly be seen that the emulsions are green and transparent, the viscosity is moderate, and the dry powder is milky and delicate, which preliminarily indicates that the preparation method and process were successful. Figure 3c shows the dispersion of the emulsion under an optical microscopy (OM). It can be seen that the surface of the microcapsule obtained after mixing the water phase with oil is smooth and has no adhesion. The uniform distribution in the emulsion is attributed to the use of a gelatin and PVA compound emulsifier. The deposition rate of the polymer on the wall slowed down, forming a smooth-surfaced microcapsule. The microcapsule particle size is about 152 μm, as shown in Figure 3d. The inner capsule wall, outer capsule wall, and hollow of the

microcapsule are clearly seen in Figure 3e. Figure 3f shows the three-dimensional structure of the microcapsule measured under 3D confocal microscope. It can be seen that the outer capsule wall is locally depressed, indicating that the interior is hollow. The reason for the depression is that the outer capsule wall is resistant to friction and has toughness, which can undergo some deformation. As can be seen in Figure 3d–f, this is consistent with the results from scanning electron microscopy (SEM) and the laser particle size tester tests.

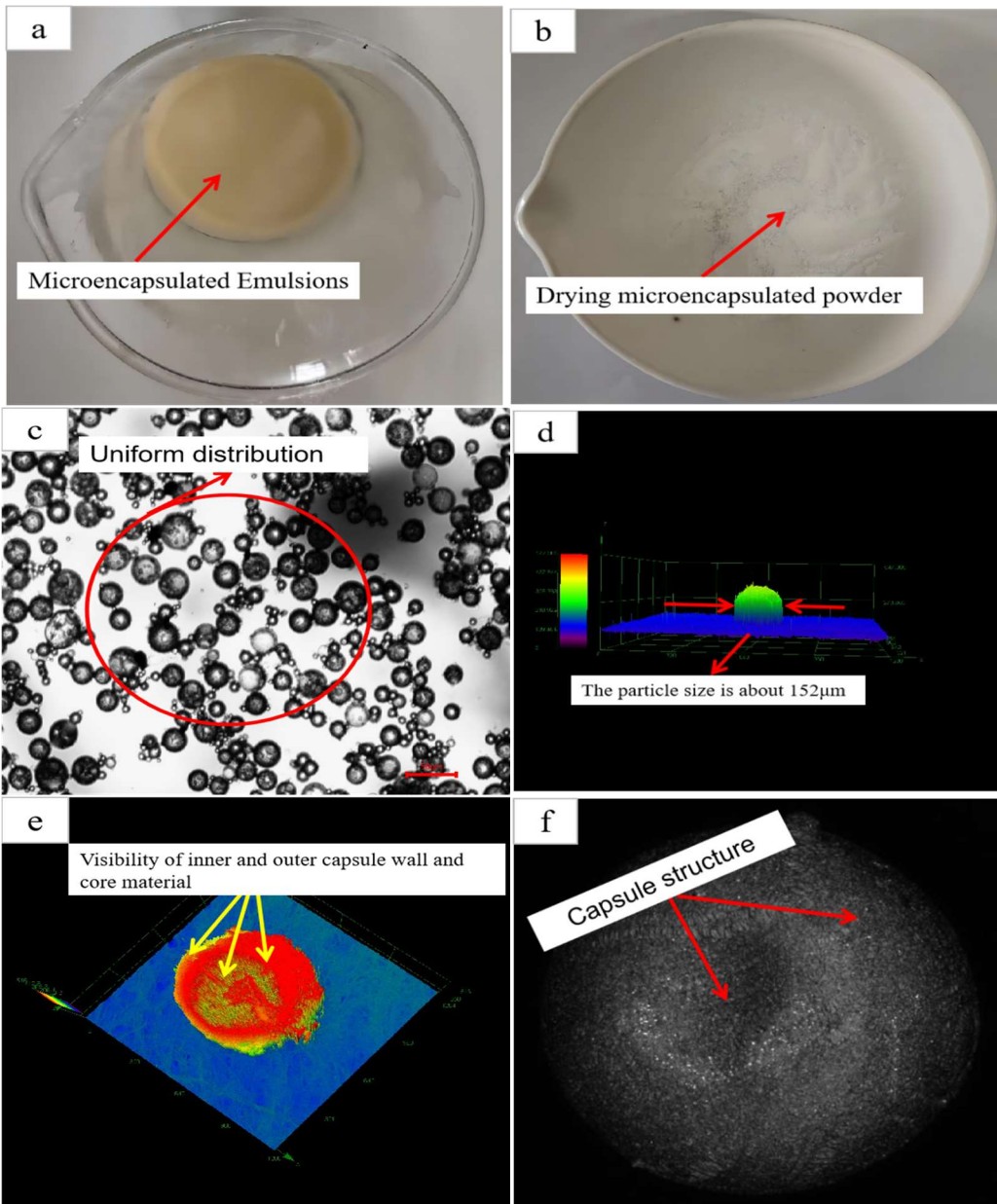

**Figure 3.** Macrostructure of prepared microcapsule and particle size under core–shell structure: (**a**) optimal emulsion; (**b**) microcapsule powder; (**c**) emulsion dispersion under optical microscope (OM); (**d**) particle size under 3D confocal microscope (LCSM); (**e**) inner capsule wall under 3D confocal microscope (LCSM); (**f**) capsule structure under 3D confocal microscope (LCSM).

### 3.1.3. Chemical Structure of Microcapsules

Figure 4a shows the FTIR spectrum of ethyl cellulose in the wall material. In the above peaks of organic functional groups, the tensile vibration of C-C aromatic ring ranges from 1586 to 1467 cm$^{-1}$. The characteristic peak corresponding to the microcapsules in Figure 4c is 1439 cm$^{-1}$. Meanwhile, the O-H-O asymmetric tensile vibration ranges from 1316 to

$1289 \text{ cm}^{-1}$. The peak value of C-O-C stretching vibration is $1227 \text{ cm}^{-1}$. This corresponds to microcapsules at the characteristic peak of $1240 \text{ cm}^{-1}$. Figure 4b shows the FTIR spectrum of glycerol. There is a stretching vibration at the $3248 \text{ cm}^{-1}$ peak for the -OH bond and an asymmetric stretching vibration at the $2913 \text{ cm}^{-1}$ peak for the $-CH_2$ bond. There is a symmetric stretching vibration for the $-CH_2$ at the $2867 \text{ cm}^{-1}$ peak and a bending vibration for the $-CH_2$ bond at the $1407 \text{ cm}^{-1}$ peak. The symmetric stretching vibration peak of C-O ranges from 1132 to $1030 \text{ cm}^{-1}$, and that of the secondary alcohol is at $839 \text{ cm}^{-1}$. These peaks coincide with those of the microcapsule at $2964 \text{ cm}^{-1}$. The results showed that the characteristic peaks of ethyl cellulose and glycerol reacted in microcapsules. This also confirmed that glycerol was successfully encapsulated in the microcapsules.

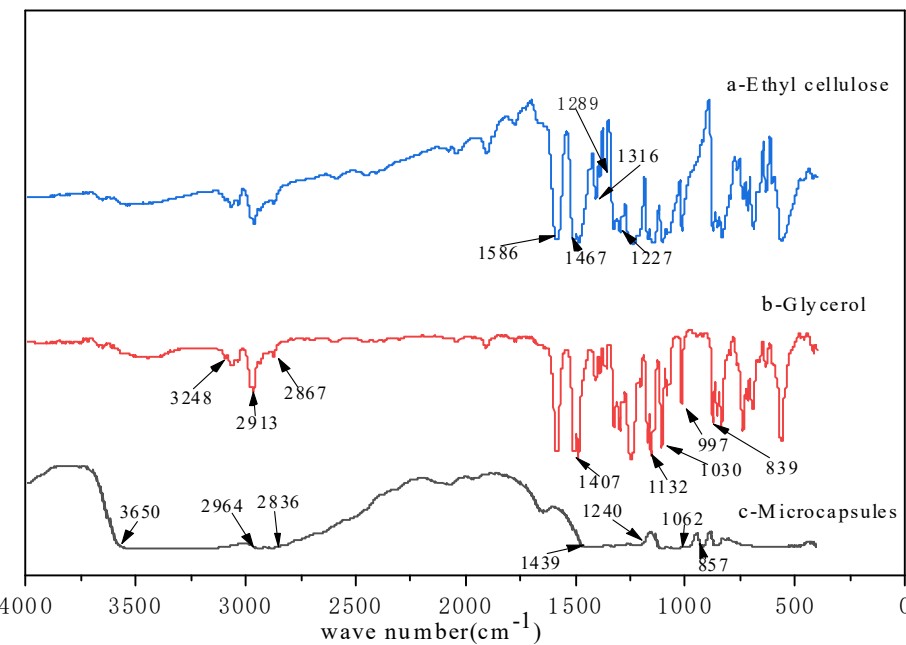

**Figure 4.** FTIR spectra of microcapsules, glycerol and ethyl cellulose: (**a**) ethyl cellulose; (**b**) glycerol; (**c**) microcapsules.

3.1.4. Conversion Efficiency of TGA and Microcapsules

Figure 5a,b show the heat weight loss curve and the conversion rate curve of microcapsules, respectively, and the initial decomposition temperature of ethyl cellulose is shown in Figure 5a at 465 °C. It has a higher initial decomposition temperature than the wood carbon structure, and the residue content is 29 wt.% at 800 °C, which easily forms a heat-stable carbon material. This is inherently because of the presence of the aromatic hydrocarbon skeleton structure in ethyl cellulose. Therefore, the initial decomposition temperature of glycerol is 150 °C, and the final decomposition temperature is 284 °C. The initial thermal degradation temperature of the microcapsules is 360 °C. It can be seen from the figure that the slope of the microcapsule curve is smaller than that of the ethyl cellulose curve. Combined with the Fourier infrared spectroscopy in Figure 4, it can be seen that glycerol is well encapsulated into the microcapsules. In Figure 5b, the thermogravimetric analysis of the figure shows that the microcapsule structure is synthesized by glycerol and the wall material. It can be seen from the TGA diagram of the microcapsules that the encapsulation rate of the microcapsules is 12 wt.%. This further indicates that the preparation of microcapsules was successful.

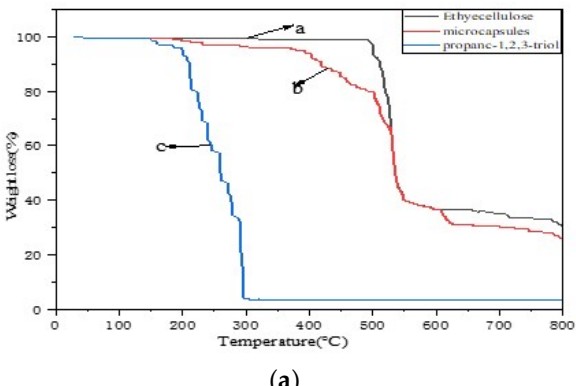
(**a**)

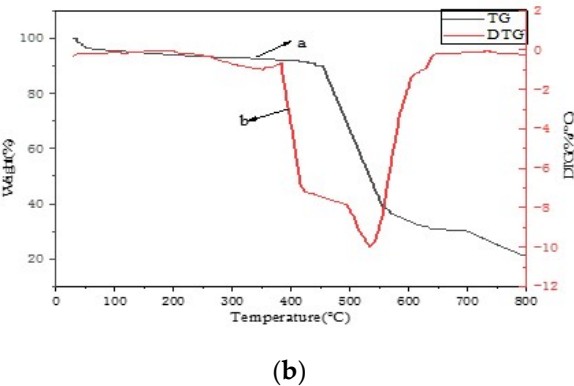
(**b**)

**Figure 5.** Fourier infrared spectra (TGA) plots of self-healing microcapsules: (**a**) a, ethyl cellulose; b, microcapsules; c, propanetriol; (**b**) a, TG; b, DTG.

### 3.2. Characterization of Optimal Core-to-Wall Ratio of Microcapsules and Corrosion Resistance of Self-Healing Coatings

Figure 6 shows the optimal core–wall ratio and corrosion resistance test of self-healing coatings after the microcapsules with different mass fractions are soaked. In Figure 6a, when the core–wall ratio of microcapsules is 1:1, the proportion of spherical microcapsules with regular morphology is low in the total number of microcapsules. The surface is not polished, and there are more impurities attached. Due to the low core material ratio, the microcapsules adhere to each other, causing excess wall polymers to accumulate on the already formed microcapsules. In Figure 6b, when the core-to-wall ratio of 1.2:1, it is easily seen that the microcapsules are uniformly dispersed. The spherical structure of the microcapsule formed is smooth and more regular, and the surface is not adhered to anything. Figure 6c When the core–wall ratio is 1.5:1, the wall content is too low. During the mixing process, the microcapsule capsule is easy to break due to its thin wall shell. Therefore, when the mass ratio of core material to wall material is 1.2:1, the microcapsules synthesized with high sphericity have a regular shape and clear structure. There is no adhesion between microcapsules. The best results were prepared at this ratio. In Figure 6d–f, the scratch corrosion morphologies of magnesium alloys soaked in non NaCl solution, 3.5% NaCl solution for 24 h and 3.5% NaCl solution for 48 h, respectively, can be seed. The analysis showed that the corrosion of the epoxy coating without microcapsules was more significant than that of epoxy coating with microcapsules. And relative to the epoxy coatings with 5% and 20%, the epoxy coating with 10% microcapsules has good corrosion resistance. This indicates that there is an optimal concentration range for self-healing microcapsules to achieve maximum corrosion resistance. Excessive microcapsule content may compromise the water repellency of the coating, resulting in reduced corrosion resistance.

### 3.3. Characterization of Self-Healing Ability of Coating by Microcapsules with Different Mass Fraction Addition

Figure 7 shows the corrosion effect of the coated samples with different amounts of microcapsules in artificial seawater. For pure epoxy resins without microcapsules. With the increase in soaking time, corrosion appeared in the scratch area, and the corrosion outside the scratch was more obvious. As can be seen from figure b, when the microcapsule content is 5%, compared to the pure epoxy coating, the scratches become shallower and lighter under the same soaking time. This indicates an initial ability to self-healing, and when the microcapsule content is 10%, only a small amount of corrosion occurs around the coating. This indicates that the microcapsules will release the core material, propanetriol, over time when the coating is scratched or broken. However, glycerol is a polar molecule, and the unshared electron pair of the oxygen element on the polar group combines with the hydrogen ion. This causes cations to adsorb on the substrate surface, which changes the structure of the double electric layer on the surface of the substrate, producing a covering

effect, forming a hydrophobic coating and weakening the movement of hydrogen ions, chloride ions and water molecules. In turn, this weakens the access to the solution substrate and hinders charge transfer. Moreover, anode polarization occurs, thereby reducing the corrosion rate and self-healing, and hindering further corrosion. When the microcapsules content in d is 20%, the ratio between microcapsules and epoxy resin becomes larger. As a result, the epoxy resin inside the microcapsule cannot be evenly dispersed and cannot be scratched at the scratch site, and the corrosion resistance is weakened as the soaking time increases. It can be seen that when the microcapsule content is 10%, the self-healing ability is the best, which is consistent with Figure 6.

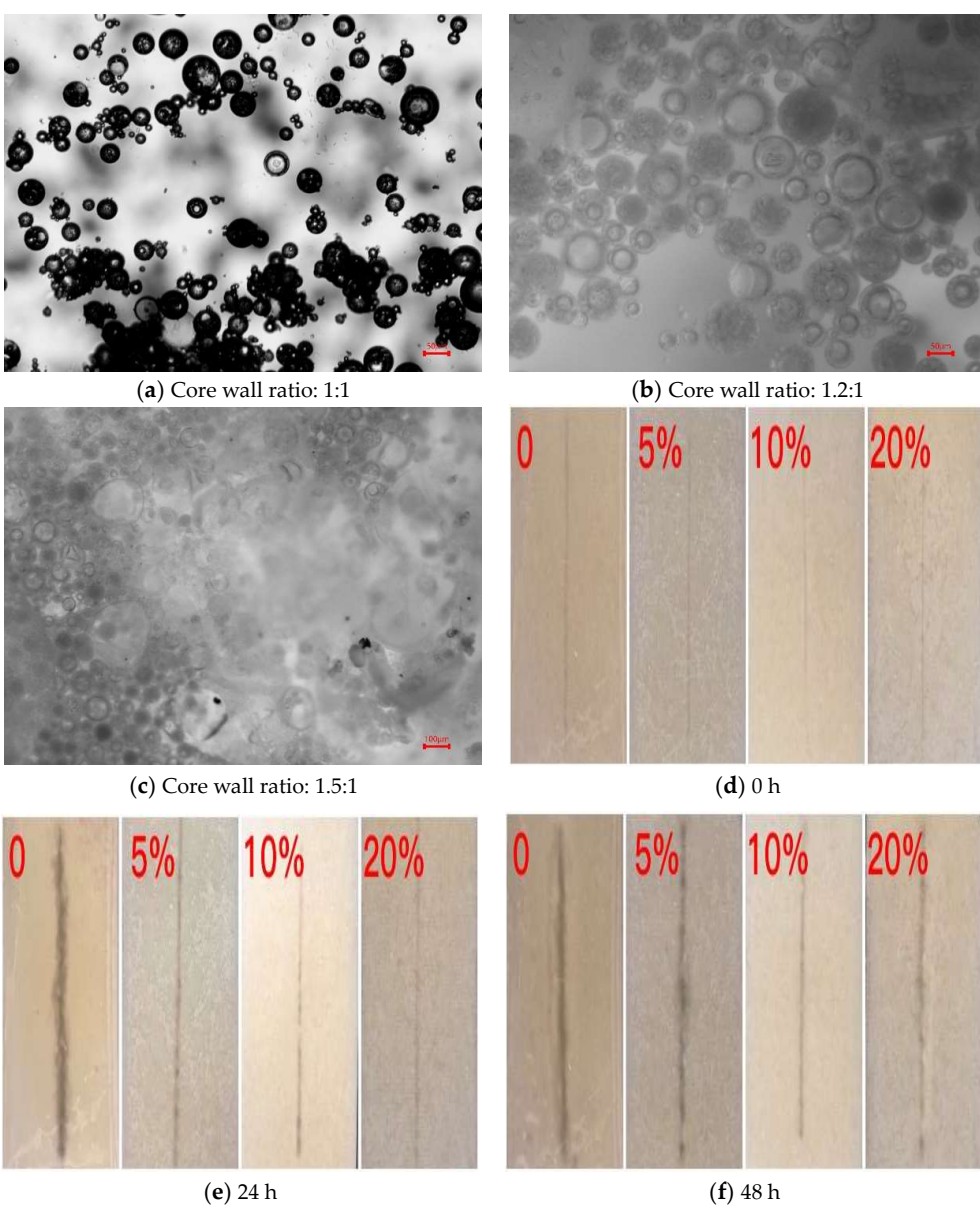

**Figure 6.** In the impregnation corrosion resistance test, the best core–wall ratio of microcapsules and self-healing coatings with different mass fractions are the following: (**a**) core-to-wall ratio of 1.5:1; (**b**) core-to-wall ratio of 1.2:1; (**c**) core-to-wall ratio of 1.5:1; (**d**) no sodium chloride solution immersion; (**e**) 3.5% sodium chloride solution immersion for 24 h; (**f**) 3.5% sodium chloride solution immersion for 48 h.

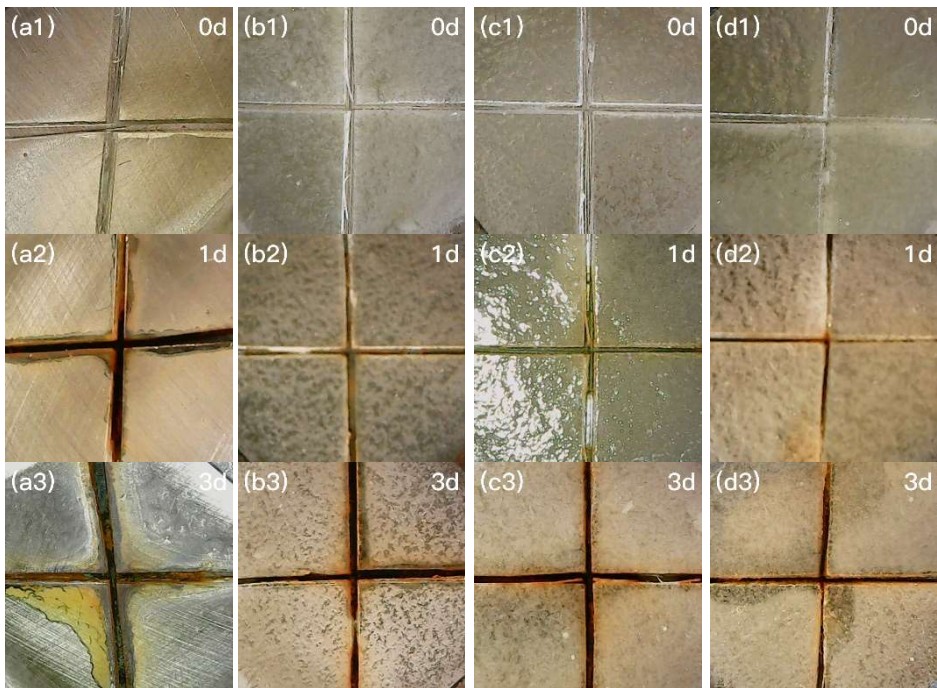

**Figure 7.** Corrosion effect of coatings with different self-healing microcapsule additions in artificial seawater: (**a**) no microcapsule addition (**a1** 0d, **a2** 1d, **a3** 3d); (**b**) microcapsule content 5% (**b1** 0d, **b2** 1d, **b3** 3d); (**c**) microcapsule content 10% (**c1** 0d, **c2** 1d, **c3** 3d); (**d**) microcapsule content 20% (**d1** 0d, **d2** 1d, **d3** 3d).

### 3.4. Electrochemical Test Characterization of Self-Healing Coatings

3.4.1. Electrochemical Impedance Spectroscopy

Figure 8 above shows the electrochemical impedance spectra of different self-healing microcapsules added to the 3.5% NaCl solution. Figure 8a shows the Nyquist diagram. The impedance modulus values in the low frequency region are $2.8 \times 10^4$, $3.6 \times 10^4$, $7.8 \times 10^4$, and $5.3 \times 10^4$, respectively. As the radius of the capacitive resistance of the capacitive arc in solution gradually increases, the corresponding resistance of the solvation layer also increases. The charge transfer resistance becomes more pronounced, which in turn hinders further erosion of the coating by the solute in solution. Moreover, when the microcapsule addition amount is 10 wt.%. The coating has the best self-healing ability, indicating that it has the best corrosion resistance at this moment. Compared to the pure epoxy coating without the addition of microcapsules, its corrosion resistance is poor. Also in the Byrd plot in Figure 8b, it can be seen that the |Z| value increases continuously in the absence of the microcapsules up until they are added up at 10 wt.%, but the value decreases at a microcapsule content of 20 wt.%. This indicates that the self-healing coating exhibits optimal impedance coating volume with a stable passive response at 10 wt.% microcapsule content [40]. The self-healing epoxy coating successfully prevented the electrolyte solution from penetrating into the substrate and coating surface. The corrosion inhibitor released by the microcapsule is adsorbed at the corrosion site. It successfully inhibits the corrosion process. The appearance of double peaks can clearly be observed in Figure 8c, indicating that the self-healing coating was successfully coated at this time and has a double-layer structure. This indicates that the corrosion resistance of the coating is optimal when the microcapsule content is 10 wt.%. In Figure 9, the error between the data fitted in the figure and the experimental data is minimal, so the equivalent circuit can be used to fit the coating data of the optimal microcapsule addition amount, where CR and CP are, respectively, the annual corrosion rate and anti-corrosion efficiency on the Tafel curve. Ecorr mainly describes the corrosion thermodynamic trend of the coating, and Icorr is the corrosion current density. In general, the main criterion of corrosion resistance is determined by

the corrosion current density. The lower the current density, the lower the corrosion rate accordingly. Epoxy resin varnish coatings and self-healing coatings have orders of magnitude lower current densities compared to magnesium alloy substrates. In addition, the self-healing coating worked better in the sodium chloride solution. The reason is that the coating successfully resists the penetration of the electrolyte solution to the substrate and the coating surface. The corrosion inhibitor released by the microcapsule is adsorbed at the corrosion site. The corrosion process was successfully suppressed, which is also consistent with the polarization curve analysis in Figure 10 below.

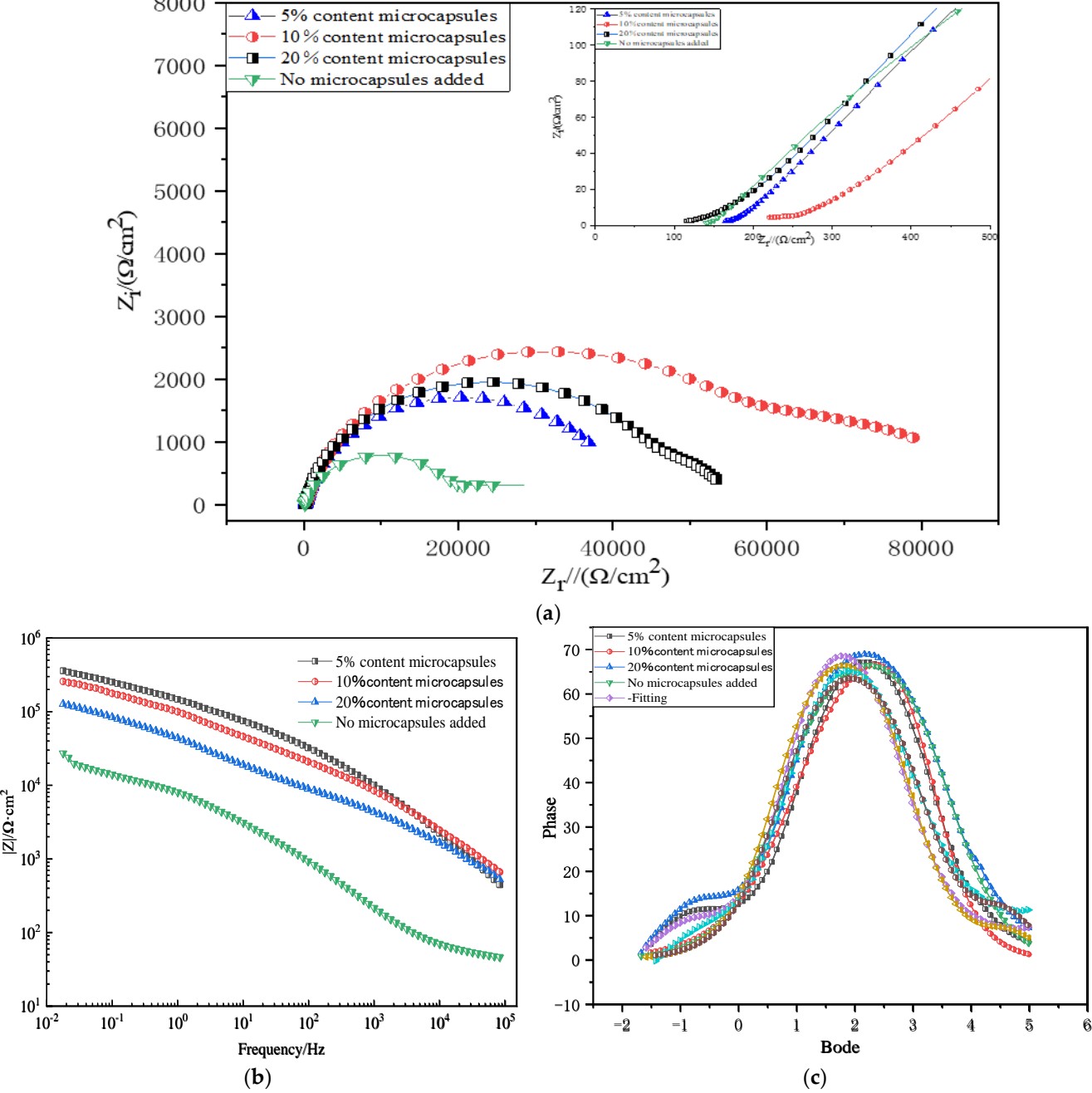

**Figure 8.** Electrochemical impedance spectra of coatings with different self-healing microcapsule additions in 3.5% NaCl solution: (**a**) Nyquist plot; (**b**) impedance vs. frequency Byrd plot; (**c**) Bode diagram of phase angle and frequency.

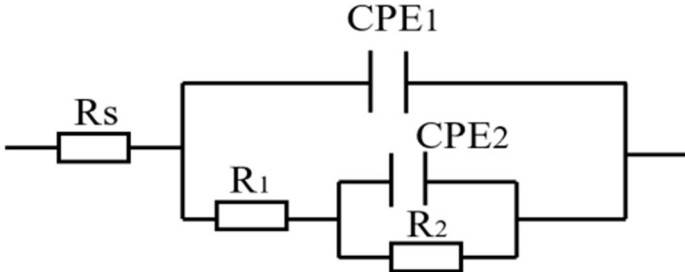

**Figure 9.** Equivalent circuit diagram of epoxy resin coating in 3.5% NaCl solution at optimal microcapsule addition.

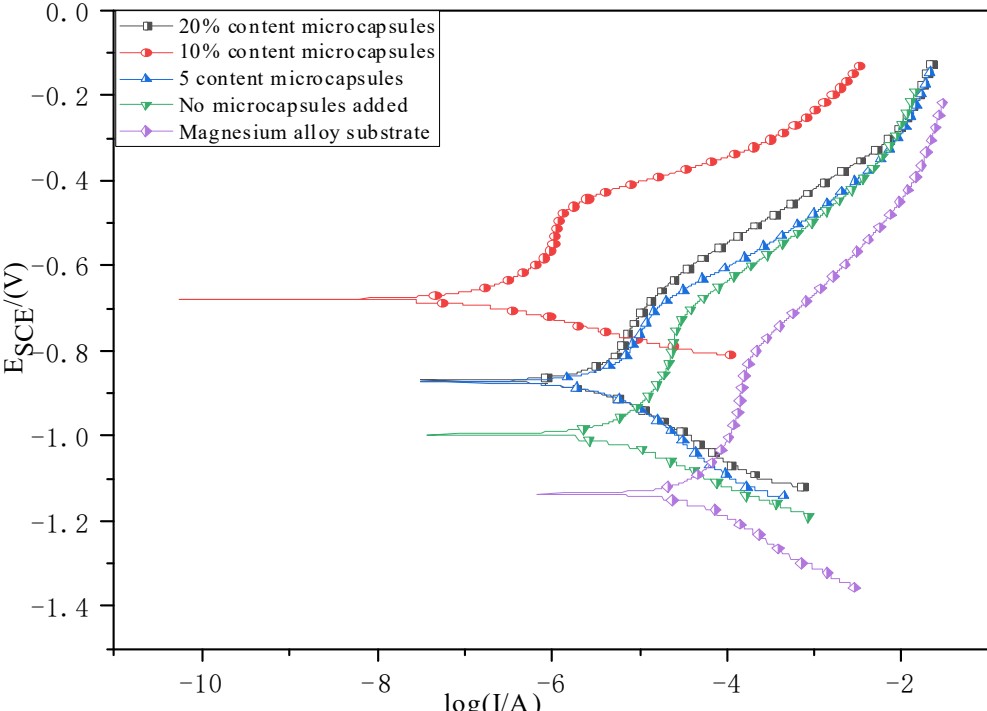

**Figure 10.** Polarization curve of epoxy resin coating in 3.5% NaCl solution at optimal microcapsule addition.

### 3.4.2. Electrochemical Polarization Curve Graph Analysis

Figure 10 shows the polarization curves of epoxy resin coatings in 3.5% NaCl solution at the optimal microcapsule dosage. Table 1 shows the Ecorr and icorr of the epoxy coating with different microcapsules added in 3.5% NaCl solution. Each set of experiments was conducted three times in parallel to exclude accidental errors. It can be concluded from the figure that when the addition of microcapsules reaches 20%, the anode arc can be observed. The corrosion potential and corrosion current density of magnesium alloy in a bare matrix are the worst, at $-1.125$ V and $1.562 \times 10^{-5}$ A, respectively. With the addition of epoxy resin varnish, the corrosion potential and corrosion current density showed an obvious optimization trend. At this time, the corrosion potential and the corrosion current density are $-1.014$ V and $1.854 \times 10^{-5}$ A, respectively. After that, according to the percentage content of microcapsule additives, the added amounts were 5 wt.%, 10 wt.% and 20 wt.%. The corresponding electrochemical measurement data showed a trend of first increasing and then decreasing, and reached the peak value when the microcapsule addition amount was 10 wt.%; that is, the optimal value was reached. The optimal corrosion potential and corrosion current density are $-0.721$ V and $1.86 \times 10^{-7}$ A, respectively. In general, the corrosion resistance of self-healing coatings is mainly determined by the corrosion current density. The lower the corrosion current density, the lower the corresponding corrosion

rate of the self-healing coating. The self-healing coating without added microcapsules has a lower order of magnitude current density compared to the added microcapsules. This indicates that both coatings can provide corrosion protection to magnesium alloy substrates and slow down the corrosion rate. However, the self-healing coating is effective because it can release microcapsules, causing the slow-release healing of coating ruptures and adsorption on the broken parts, which can inhibit the corrosion process well. Table 2 shows the equivalent fitting of impedance using electrochemical fitting software, and the fitted equivalent circuit diagram of the coating is shown in Figure 9, which shows that the fitted data and the experimental data are similar. The result error is small, so the equivalent circuit diagram can be used to fit the experimental data. Rs represents the resistance of the solution, CPE1 and CPE2 represent the capacitance and resistance of the self-healing coating, and R1 and R2 represent the charge transfer resistance between the coating and the reaction resistance of the coating, respectively. Due to the rough surface and inhomogeneous electrochemical properties of the magnesium alloy substrate, a phase angle element is commonly used instead of the capacitance for interpreting the behavior of the high-frequency capacitive arcs. CPE1 and CPE2 represent the phase angle elements of the reaction interface and the two-layer corrosion products, respectively. The results show that the higher the fitting resistance of microcapsules, the better the corrosion resistance. When the mass fraction of the microcapsule is 10 wt.%, its corrosion resistance reaches its maximum. The results are consistent with those of the polarization curve.

**Table 1.** Ecorr and icorr of epoxy resin coatings with different microcapsule additions in 3.5% NaCl solution.

| Microcapsules | $E_{corr}$/V | $i_{corr}$/(A/cm$^2$) |
|---|---|---|
| Uncoated | −1.125 | $1.562 \times 10^{-5}$ |
| Pure epoxy resin coating | −1.014 | $1.854 \times 10^{-5}$ |
| 5 wt.% epoxy-based coating of microcapsules | −0.931 | $1.538 \times 10^{-6}$ |
| Epoxy-based coating of 10 wt.% microcapsules | −0.721 | $1.86 \times 10^{-7}$ |
| Epoxy-based coating of 20 wt.% microcapsules | −0.903 | $1.507 \times 10^{-6}$ |

**Table 2.** Tafel curve extrapolation results of epoxy resin coatings with different microcapsule additions in 3.5% NaCl solution.

| Microcapsules | $R_s$ $\Omega \cdot cm^2$ | $CPE_1$ $\Omega^{-1} \cdot S^{-n} \cdot cm^2$ | $n_1$ | $R_1$ $\Omega \cdot cm^2$ | $CPE_2$ $\Omega^{-1} \cdot S^{-n} \cdot cm^2$ | $n_2$ | $R_2$ $\Omega \cdot cm^2$ | $R_1 + R_2$ $\Omega \cdot cm^2$ |
|---|---|---|---|---|---|---|---|---|
| Uncoated | 19.92 | $8.32 \times 10^{-6}$ | 0.87 | 8110 | $4.8 \times 10^{-4}$ | 0.61 | 15,890 | 24,000 |
| Pure epoxy resin coating | 12.13 | $8.94 \times 10^{-6}$ | 0.89 | 26,450 | $2.3 \times 10^{-4}$ | 0.85 | 13,150 | 39,600 |
| 5 wt.% epoxy-based coating of microcapsules | 11.2 | $8.44 \times 10^{-6}$ | 0.89 | 14,080 | $1.7 \times 10^{-4}$ | 0.73 | 39,820 | 53,900 |
| Epoxy-based coating of 10 wt.% microcapsules | 9.8 | $1.33 \times 10^{-5}$ | 0.82 | 13,520 | $2.3 \times 10^{-4}$ | 0.85 | 68,900 | 82,420 |
| Epoxy-based coating of 20 wt.% microcapsules | 13.08 | $2.49 \times 10^{-5}$ | 0.84 | 28,900 | $4.4 \times 10^{-4}$ | 0.62 | 24,180 | 53,080 |

### 3.4.3. SEM Image and EDS Image of Self-Healing Coating

Figure 11 shows the planar SEM image, sectional SEM image and the corresponding sectional EDS spectra of the epoxy resin coating with optimal microcapsule dosage. At the magnification of Figure 11a, 200 μm, it can be seen that the encapsulated small particles are uniformly dispersed on the surface of the self-healing coating. This indicates that the microcapsules at the optimal content are well formed by stirring and curing after being added to the epoxy resin coating. When the coating is damaged by external forces, the microcapsules encapsulated in the coating can flow out in time to release the retarding agent, glycerin. In turn, the coating provides good protection against corrosion. Figure 11b shows the enlarged view in a. At this time, under the amplification size of 100 μm, the structure of the microcapsule can be seen increasingly clearly, and the dispersion is very

good. Figure 11c shows the cross-sectional SEM image of the epoxy resin coating at the optimal microcapsule dosage. Figure 11d shows the EDS energy spectrum of the corresponding cross section in Figure 11c, which corresponds to the overall face-swept element distribution. The thickness of the self-healing coating is 95–100 μm. It is clearly seen that the incorporation of microcapsules results in a tighter adhesion of the coating to the magnesium alloy substrate and thus excellent corrosion resistance.

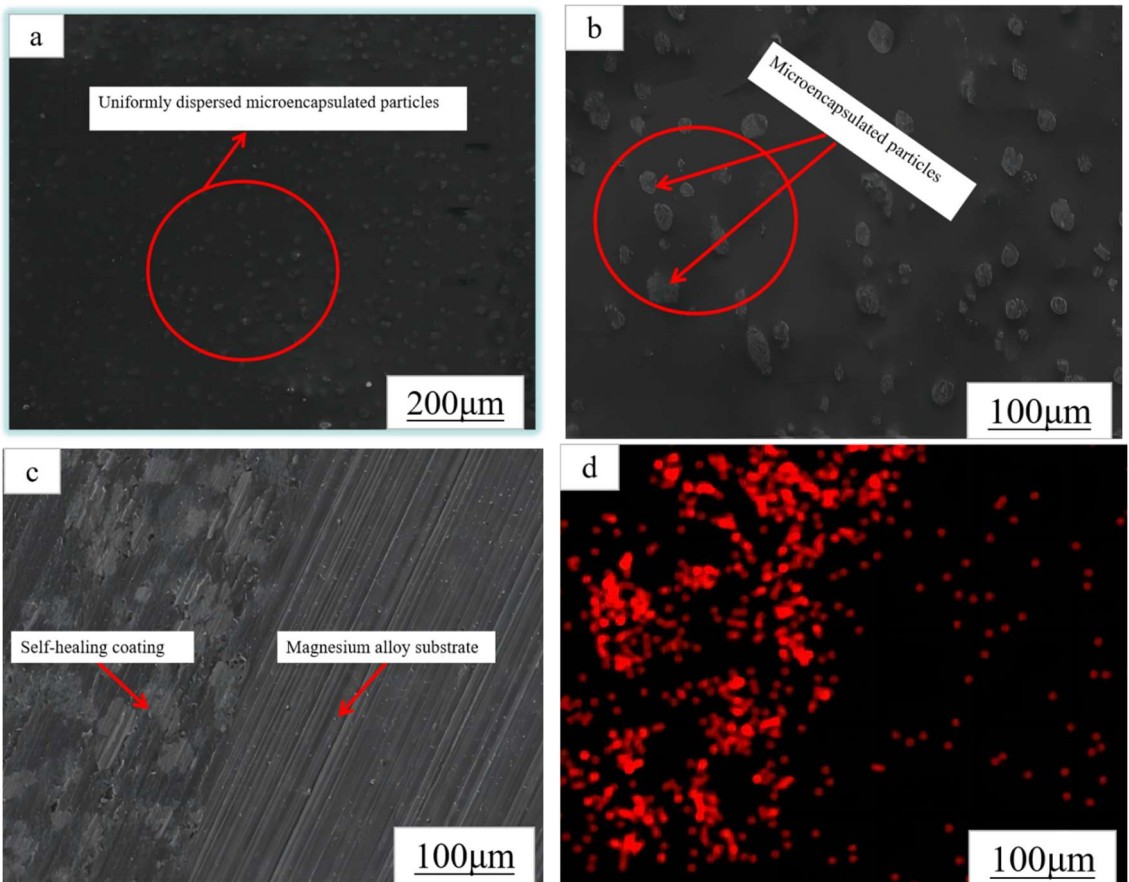

**Figure 11.** Plane SEM and cross-sectional SEM and EDS spectra of epoxy resin coatings at optimal microcapsule addition levels: (**a**) 200 micron SEM; (**b**) 100 micron SEM; (**c**) cross section of the self-healing coating was examined by SEM at 100 microns; (**d**) EDS spectra of self-healing coating cross section at 100 microns.

## 4. Conclusions

The present work is based on the preparation of loaded propanetriol microcapsules in micrometer vessels. Because ethyl cellulose is easily decomposed by heat, it is biodegradable and chemically stable. Glycerol itself, due to its lubricating and oiliness, can form a good hydrophobic coating with hydrogen ions on the surface of magnesium alloys. In turn, it can slow down hydrogen ions, chloride ions and water molecules from penetrating the epoxy resin coating, which can effectively reduce the corrosion resistance of the coating and enhance the self-healing performance of the coating, specifically as follows:

1.  Microcapsules containing the corrosion inhibitor propanetriol are added into the epoxy resin to form a round and smooth spherical structure, which is uniformly dispersed into the coating.
2.  The particle size of the propanetriol-loaded microcapsules was (150.424 ± 3.756) μm as observed by SEM, laser particle size tester, and 3D confocal microscopy.
3.  The results of TGA and FTIR show that propanetriol, as a corrosion inhibitor, was effectively encapsulated inside the capsule wall as a core material for release retardation

when rupturing of the microcapsules occurred. The DTG curves of the microcapsules show that the capsule wall shell are loaded with about 12% of the printing agent.

4. SEM images of the plane and cross-section of the self-healing coating show that the microcapsules are evenly distributed in the epoxy resin layer, and the incorporation of microcapsules makes the coating adhere more closely to the magnesium alloy substrate.

5. The electrochemical experimental analysis shows that the microcapsules are successfully added into the epoxy resin coating. Glycerol, as a corrosion inhibitor, is able to be successfully released. It is adsorbed to the rupture site to form a hydrophobic coating layer for self-healing, which in turn attenuates the corrosion rate of the magnesium alloy.

**Author Contributions:** Conceptualization, L.L. and S.Z.; methodology, L.L.; software, L.L.; validation, S.Z., Q.L. and L.L.; formal analysis, S.Z.; investigation, L.L.; resources, S.Z.; data curation, L.L.; writing—original draft preparation, L.L.; writing—review and editing, L.L.; visualization, J.B. and S.Z.; supervision, Y.X.; project administration, T.Z. and S.Z.; funding acquisition, S.Z and T.Z. All authors have read and agreed to the published version of the manuscript.

**Funding:** This work was supported by the key project of National Natural Science Foundation of China: Research basis of "corrosion-functional" integrated protective coating on magnesium alloys (U21A2045), the National Natural Science Foundation of China (NSFC), "Study of conductive-corrosion resistant chemical conversion coating of magnesium alloy" (51771050), and the Basic Research Project of Liaoning Provincial Education Department (LJKMZ20220600).

**Institutional Review Board Statement:** Not applicable.

**Informed Consent Statement:** Not applicable.

**Data Availability Statement:** Not applicable.

**Conflicts of Interest:** The authors declare no conflict of interest.

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
