# Peer review of "Research on the Corrosion Resistance of an Epoxy Resin-Based Self-Healing Propylene Glycol-Loaded Ethyl Cellulose Microcapsule Coating"

_coatings, doi:10.3390/coatings13091514_

Round 1
Reviewer 1 Report
Present manuscript indicated that the propanetriol was successfully released from the broken microcapsules for self-healing, forming a magnesium alloy anticorrosion coating with excellent corrosion resistance and self-healing ability. Some comments given as follows:
1. Line 34, Nothing truly unique in its current state. Because of the lack of a novel, the current submission looks to be a replication or modified work. The authors must describe their novel in detail. This work should be rejected owing to a major issue.
2. Line 35-36, related to magnesium alloys, please give additional relevant reference as follows: https://doi.org/10.3390/ma16093298
3. Line 41-42, rather than informing the specific scope from China, would be recommended to wider scope like in the world trend.
4. Line 43, the explanation of expansion needs more specific.
5. Line 91, the authors objective to studied polyvinyl alcohol and gelatin, but nothing any explanation for why should use both of them. Please give it in the Introduction section first.
6. Line 139-140, why the materials preparation is performed like that for AZ91D, it is a golden ration of have a specific guideline? Explain it.
7. Line 165 the addition of 5%, 10% and 20% as a variation in this parameter needs further explanation, of it is adopted from the previous literature?
-
Author Response
Dear reviewer:
First of all, thank you very much for taking time out of your busy schedule to give valuable comments and suggestions on this paper, for each of your questions and suggestions. I will seriously reply and adopt, and in the original text I used red to modify the mark, so as to facilitate your review of the manuscript to check, the following is my answer to each question, please go through it:
-
Dear reviewers, Thank you for your comments. I have made the corrections to the lack of novel in the introduction. The experimental procedure and the novelty were expressed by organizing and writing the manuscript. Compared with the existing literature, the unique feature is that ethylcellulose itself has the functions of bonding, filling, and film-forming. Previous research work has rarely focused on emulsifiers under compounding. Commonly used is the analysis of the morphology of the microcapsules under a single emulsifier and the lack of a certain core-to-wall ratio and an optimal mass fraction between the whole and the coating. This is the focus of the work. It can be shown through experiments that it can repair the gaps created by the coating itself very well, further refine the surface of the substrate, and have a better contribution to the corrosion resistance. At the same time, in the last paragraph of the introduction, the purpose of this experimental research, as well as the process of investigation, I have carried out a certain amount of analysis and description, so that it looks more detailed, read more clearly and accurately expressed.
-
Dear reviewer, thank you for your question . Regarding the content related to magnesium alloys in lines 35-36, I apologize for not expressing specificity during the description of my manuscript. I have made the relevant changes according to your request. I have further improved the description of the application areas and functions of magnesium alloys.
-
Dear Reviewer, First of all, thank you for your comments and details . It is a good I need to improve the area . Below I have further improved the content of the paper according to your comments. From the original "With the beginning of 2025 China's intelligent manufacturing" to the current "However, China is a large magnesium resource consumption country, in today's world development trend, it is not difficult to predict, magnesium alloy further in-depth research and application will be more significant".
-
Dear reviewer, thank you very much for pointing out the specific schema of line 43 in the paper. After carefully reading through and analyzing it, I will delete and modify it to read: "Therefore, in this context, magnesium alloy itself has poor corrosion resistance, which restricts the development of magnesium alloy application surface, this status quo, urgently needs to be solved". Make the context look more coherent and persuasive. At the same time, thank you again for your valuable questions, I will receive suggestions with an open mind, actively adopt them, and seriously modify them to make the paper look more rigorous and professional.
-
Dear reviewers, Thank you very much for your valuable suggestions . Regarding the use of polyvinyl alcohol and gelatin, I will give it in the introduction section as you suggested. I sincerely apologize for not taking this into consideration. I have carefully read the additions in the introduction section. The details are as follows: The formation of emulsion droplets involves both dispersion and stabilization. Therefore, sometimes when an emulsifier is used alone, the own viscosity of the aqueous phase is too low and the droplets in the emulsion are highly susceptible to agglomeration. In this context, polyvinyl alcohol itself has bonding, emulsifying and dispersing properties . Gelatin itself is a macromolecular hydrophilic colloid, which can also be used for bonding. Therefore, the simultaneous addition of polyvinyl alcohol and gelatin can play a stabilizing role as well as enhance the viscosity of the aqueous phase and form a thin film, thus improving the encapsulation rate of microcapsules. Whereas, if emulsifiers and stabilizers are used improperly or in insufficient amounts, they may be reversed, releasing the core and leading to a significant decrease in microencapsulation encapsulation rate. Finally, thank you again for your valuable comments and suggestions, thank you.
-
Dear reviewers, thank you for your comments . I will explain in detail below, the main reason for the following two points: â‘ Because this experiment is based on the research direction of my subject set up, and my main research is the preparation of AZ91D magnesium alloy surface composite coating and performance research. Therefore, the first choice is AZ91D magnesium alloy; â‘¡ There is no specific guideline for the golden ratio, based on a large number of references and then experimental results program.
-
Dear Reviewer, Thank you for your comment. I will carefully respond in detail to your questions below . Increasing the percentage of microencapsulation mass fraction from no addition of varnish, to addition of 5%, 10% and 20% is not what was used in the previous literature. Rather, I have done a lot of one-way experiments in the early stage and then came up with the optimal quality ratio. This is the way to set the added mass percentage, and thus the coating self-repair gradient situation is the most obvious, the easiest to find the best results.

Reviewer 2 Report
The authors produced microcapsules with synergistic self-healing effect with slow release by solvent evaporation method. The objective was to show that the self-healing layer with added microcapsules has self-healing properties, corrosion resistance characteristics, adhesion and tensile ductility with self-healing under the action of the emulsifier mixture. Several morphological and chemical/electrochemical characterization techniques were used.
The subject of the manuscript is interesting since it seeks to solve the problem of the poor corrosion resistance of the magnesium alloy itself.
The manuscript requires major revision before it can be considered for publication. The following aspects should be reviewed:
English should be revised throughout the manuscript. In some paragraphs, strange expressions even appear, as if they were translated by AI that does not understand what it is about.
Title
Self-healing epoxy resin-based ethyl cellulose microcapsule coating was tested for magnesium surface. I think this should appear in the title.
Abstract
It presents a lot of information (which seems to be repeated in the last paragraph of the Introduction). The final part about the impedance results should be reformulated because it is not clear what the authors wanted to say.
1. Introduction
- Authors should present in more detail and quantifiable what innovation they have brought and what are the advantages of their work developed in this paper compared to others.
3. Results and Discussion
3.3. Electrochemical test characterization of self-healing coatings
3.3.1. Electrochemical impedance spectroscopy
- The authors stated that “Figure 8(a) shows the Nyquist plot from which it is viewed. The impedance modulus values in the low frequency region are 2.8 × 104, 3.6 × 104, 7.8 × 104, and 5.3 × 104, respectively.” Please explain how these values were obtained? What is the unit of measure?
- The authors presented the electrical circuit used to fit the impedance spectra, but did not present the results of the fitting, how the electrical parameters vary depending on the content of the microcapsules. These parameters should be obtained and provided in a table with the corresponding explanations in the text. The authors said that these parameters should be in Table 1, but there are other parameters that are not presented/explained in the text. Please review the values in Table 1 and explain all parameters (CR, PE).
3.3.2. Graphical analysis of the electrochemical polarization curve
- Similar to section 3.3.1. Please match the graphs, tables and explanations in the text.
English should be revised throughout the manuscript. In some paragraphs, strange expressions even appear, as if they were translated by AI that does not understand what it is about.
Author Response
Dear Reviewer:
First of all, thank you very much for taking time out of your busy schedule to give your valuable comments and suggestions on this thesis. For each of your questions and suggestions, I will seriously reply and adopt them, and in the original text I have marked the changes in red, so that it is convenient for you to review the manuscript, the following is my answer to each question, please go through it:
- Dear Reviewer, Thank you for your comment . Regarding the title of the paper you mentioned, your suggestion is very helpful. I have changed the title from "Research on the corrosion resistance of an epoxy resin-based self-healing propylene glycol-loaded ethyl cellulose microcapsule coating" to "Self-healing epoxy resin-based ethyl cellulose microcapsule coating was tested for magnesium surface" to make it more useful. surface" to make it look more detailed and read more clearly and accurately.
-
Dear reviewer, Firstly, thank you very much for your comment . Secondly, I will provide a corresponding detailed response to the above questions. The electrochemical experiments conducted in this experiment to measure the impedance values were performed three times in parallel to exclude the factor of chance error. The purpose of this experiment was to treat the measured impedance results as one of the reference indicators. The final impedance value in 3.5 wt% NaCl solution was 8.242 × 104, which is the optimum value.
-
Dear reviewer, thank you for your comment. Regarding the problems you mentioned in the introduction section, I have made corresponding changes to make it look more complete. In the introduction section, a comparative analysis between this experiment and other experiments in the literature is conducted to show the novelty of this experiment. At the same time, in the last paragraph of the introduction, I have analyzed and explained the purpose of this experimental research and the process of investigation. Make it look more detailed, read more clearly and accurately expressed.
-
Dear reviewer, Thank you very much for your comment reply . I will give a detailed reply to each of your questions: in the equivalent circuit in Fig. 8(a), the values of RΩ and Rct can be obtained directly from the Nyquist plot through this equivalent circuit model, using the fitting software ZsimpWin. Also, the unit of measure of these values is Ω
-
Dear reviewer, Thank you for your comment . This is a good I need to improve the following I make a specific analysis of the chart to explain: â‘ The polarization curve calculated by Tafel extrapolation is shown in Table 1. Where CR and CP are the annual corrosion rate and corrosion protection efficiency on the Tafel curve, respectively. Ecorr mainly describes the corrosion thermodynamic trend of the coating, and is not primarily a criterion for evaluating corrosion. â‘¡ In general, the main criterion for corrosion resistance is judged by the corrosion current density. The lower the current density, the correspondingly lower the corrosion rate. Epoxy resin varnish coatings and self-healing coatings have orders of magnitude lower current densities compared to magnesium alloy substrates. In addition, the self-repair coating worked better in the sodium chloride solution solution. The reason for this is that the coating successfully resisted the penetration of the electrolyte solution into the substrate and coating surface. The microcapsules released corrosion inhibitors that adsorbed at the corrosion site. The corrosion process was successfully inhibited.â‘¢According to the experiment can be analyzed, without adding microcapsules and with the microcapsules added mass fraction percentage content changes, the corresponding electrochemical experimental parameters, such as corrosion potential and corrosion current density will change accordingly, appearing to increase first and then reduce the trend, the peak in the middle of the peak, that is, the optimal value.
-
Dear Reviewer, Thank you for your comment. In the following, I will carefully respond in detail to the questions you have raised. The description of this paragraph in the paper has been modified to read in the following form: Fig. 10 shows the polarization curves of epoxy resin coatings in 3.5% NaCl solution with optimum microcapsule additions. Table 2 shows the Ecorr and icorr of epoxy coatings with different microcapsules added in 3.5% NaCl solution. where each set of experiments was performed in parallel three times to take out the chance errors. From the graphs, it was concluded that anodic arcs were observed from no microcapsules added up to an addition of 20 wt%. The corrosion potential as well as the corrosion current density under the bare substrate of magnesium alloy is the most inferior, which is -1.125V and 1.562×10-5A, respectively; with the addition of epoxy resin varnish, the corrosion potential as well as the corrosion current density has an obvious optimization trend, at this time, the corrosion potential and the corrosion current density are -1.014V and 1.854×10-5A, respectively; after that in order with the microcapsule additive quantity percent content ratio changes, the addition amount of 5wt%, 10wt%, 20wt%. The corresponding electrochemical measurement data values appeared to increase and then decrease, and in the microcapsule additive amount of 10wt%, the peak value, that is, to reach the optimal value. The corrosion potential and corrosion current density at the optimum value were -0.721 V and 1.86×10-7 A. In general, the corrosion resistance of self-healing coatings is mainly determined by the corrosion current density. The lower the corrosion current density, the corresponding corrosion rate of the self-healing coating will be reduced. The self-repair coating without added microcapsules has a lower order of magnitude current density compared to the added microcapsules. This indicates that both coatings can provide corrosion protection to magnesium alloy substrates and slow down the corrosion rate. However, the self-repair coating is effective in that it can release microcapsules for slow-release repair of coating ruptures and adsorption on the broken parts, which can inhibit the corrosion process well. Table 3 shows the equivalent fitting of impedance using electrochemical fitting software, and the fitted equivalent circuit diagram of the coating is shown in Fig. 9, which shows that the fitted data and the experimental data are similar, and the result error is small, so the equivalent circuit diagram can be used to fit the experimental data.Rs represents the resistance of the solution, CPE1 and CPE2 represent the capacitance and resistance of the self-repair coating, and R1 and R2 represent the capacitance and resistance between the coating and the solution, R1 and R2 represent the charge transfer resistance between the coating and the reaction resistance of the coating, respectively. Due to the rough surface and inhomogeneous electrochemical properties of the magnesium alloy substrate, a phase angle element is commonly used instead of the capacitance for interpreting the behavior of the high-frequency capacitive arcs, and CPE1 and CPE2 represent the phase angle element of the corrosion products in the reactive interfaces and of the bilayer, respectively. The results show that the larger the fitted resistance, the better the corrosion resistance, with the maximum resistance at a mass fraction of 10 wt% of microcapsule addition. The results are in agreement with those obtained from polarization curves.

Round 2
Reviewer 1 Report
Reviewers greatly appreciate the efforts that have been made by the author to improve the quality of their articles after peer review. I reread the author's manuscript and further reviewed the changes made along with the responses from previous reviewers' comments. Unfortunately, the authors failed to make some of the substantial improvements they should have made making this article not of decent quality with biased, not cutting-edge updates on the research topic outlined. In addition, the author also failed to address the previous reviewer's comments, especially on comments number 1 (lack of novel), 2 (not incorporate the literature), and 3 (not addressed). Thank you very much for the opportunity to read the author's current work.
-
Reviewer 2 Report
Dear Editor,
I suggest the publication of the revised manuscript.